# Synthesis and Characterization of a Full-Length Infectious cDNA Clone of *Tomato Mottle Mosaic Virus*

**DOI:** 10.3390/v13061050

**Published:** 2021-06-01

**Authors:** Liqin Tu, Shuhua Wu, Danna Gao, Yong Liu, Yuelin Zhu, Yinghua Ji

**Affiliations:** 1College of Horticulture, Nanjing Agricultural University, Nanjing 210095, China; liqintu1212@126.com (L.T.); 2020204035@stu.njau.edu.cn (D.G.); 2Institute of Plant Protection, Jiangsu Academy of Agricultural Sciences/Key Lab of Food Quality and Safety of Jiangsu Province-State Key Laboratory Breeding Base, Nanjing 210014, China; wushuhua@jaas.ac.cn; 3Institute of Plant Protection, Hunan Academy of Agricultural Sciences, Changsha 410125, China; haoasliu@163.com

**Keywords:** *tomato mottle mosaic virus*, full-length genome sequence, infectious cDNA clone, *Agrobacterium*-mediated inoculation, infectivity

## Abstract

*Tomato mottle mosaic virus* (ToMMV) is a noteworthy virus which belongs to the Virgaviridae family and causes serious economic losses in tomato. Here, we isolated and cloned the full-length genome of a ToMMV Chinese isolate (ToMMV-LN) from a naturally infected tomato (*Solanum lycopersicum* L.). Sequence analysis showed that ToMMV-LN contains 6399 nucleotides (nts) and is most closely related to a ToMMV Mexican isolate with a sequence identity of 99.48%. Next, an infectious cDNA clone of ToMMV was constructed by a homologous recombination approach. Both the model host *N. benthamiana* and the natural hosts tomato and pepper developed severe symptoms upon agroinfiltration with pToMMV, which had a strong infectivity. Electron micrographs indicated that a large number of rigid rod-shaped ToMMV virions were observed from the agroinfiltrated *N. benthamiana* leaves. Finally, our results also confirmed that tomato plants inoculated with pToMMV led to a high infection rate of 100% in 4–5 weeks post-infiltration (wpi), while pepper plants inoculated with pToMMV led to an infection rate of 40–47% in 4–5 wpi. This is the first report of the development of a full-length infectious cDNA clone of ToMMV. We believe that this infectious clone will enable further studies of ToMMV genes function, pathogenicity and virus–host interaction.

## 1. Introduction

Tomato (*Solanum lycopersicum* L.) is widely cultivated in the world and is of great economic value among vegetable crops. In 2016, approximately 5 million hectares of tomatoes were planted globally, producing 177 million tons of tomatoes, with China and India the two largest producers [1]. At least 136 different virus species infecting tomato have been reported in the world, with 34 of these viral species reported in China, including 23 RNA viruses and 11 DNA viruses [2]. *Cucumber mosaic virus* (CMV), *tobacco mosaic virus* (TMV), *tomato spotted wilt virus* (TSWV) and *tomato yellow leaf curl virus* (TYLCV) are commonly encountered by the growers, and symptoms caused by these viruses in tomato may vary depending on the growth stage, time of infection and environmental conditions, but typically include plant stunting; yellowing, mottling, bronzing, spotting and malformation of leaves; and development of light-colored concentric rings on fruits [3,4,5,6]. Moreover, tomato viruses have caused destructive damage and serious economic losses worldwide and pose a major threat to global tomato safety production [7,8,9]. Significantly, it is important to unravel the molecular mechanisms of viral infection and develop effective control strategies for tomato viruses.

*Tobamovirus* is the largest genus within the Virgaviridae family, and viruses within this genus cause fatal damage to crops and great economic losses worldwide. ToMMV is an emerging virus belonging to the Virgaviridae family, with a genome of 6399 nucleotides comprising four open reading frames (ORFs) that encode four proteins, including the 126 K protein, 183 K protein, movement protein (MP) and coat protein (CP). ToMMV was first found in tomato in Mexico and was officially named in 2013 [10]. Since then, it has spread fast in the world, including China [11,12], America [13,14] and Spain [15]. Notably, it has been reported that the 126 K protein and the 183 K protein are the components of the RNA replication protein complex which are mainly involved in ToMMV viral RNA replication [16]. It is well known that the MP is needed for the cell-to-cell movement of *Tobamovirus* [17] and the CP is critical for symptom development, virion formation, viral replication and the long-distance movement of *Tobamovirus* [18,19,20]. However, the molecular mechanisms of the MP and CP in the movement of ToMMV remain elusive. Given that a full-length infectious cDNA clone of ToMMV has not been constructed so far, it is not surprising that the pathogenicity of ToMMV is in most cases far from being fully understood.

Infectious cDNA clones (ICs) are constructed from stable cDNA copies of viral sequences, and they are amenable viral genetic materials which can be used for consistent infection by constructing infectious clones from DNA/cDNA of viruses [21,22,23]. An increasing number of studies have reported that ICs are powerful tools to investigate the viral gene functions, disease pathology and virus–host interactions in human, animal and plant viruses [24,25,26,27]. Generally, the full-length genome cDNA is first cloned and then infectious transcripts are produced in vitro using *bacteriophage* T7 promoter and T7 RNA polymerase [28,29]. In addition, the *cauliflower mosaic virus* (CaMV) 35S promoter has also been used extensively for ICs construction since it was reported by Odell et al. in 1985 [30]. Since then, it has been used for developing full-length infectious cDNA clones of numerous viruses [31,32,33,34,35]. Significantly, numerous viruses in the genus *Tobamovirus* have been constructed in the preceding decades. Viral factors that are responsible for the resistance-breaking character of Ob, a *Tobamovirus* that overcomes the *N* gene-mediated hypersensitive response (HR), were determined by identifying a variant that induced the HR through both site-directed mutagenesis of the infectious Ob cDNA clone and mutagenesis of it with hydroxylamine [36]. Full-length cDNAs of *tomato mosaic virus* (ToMV) were cloned into dual CaMV 35S and T7 promoter-driven vectors [37,38], and this revealed that a single residue substitution could induce a systemic HR in *N. benthamiana* [38]. It has also been demonstrated that the two mutants pCGMMV-CP-D89A and pCGMMV-RdRp-E1069A can prevent the superinfection of *cucumber green mottle mosaic virus* (CGMMV) in *N. benthamiana* by inoculation with ICs constructed with the CaMV 35S promoter [39]. These reports shed light on not only the synthesis of ICs of other *Tobamovirus* but also the construction of virus-based plant expression vectors. Despite the full-length genomes of ToMMV having been cloned and reported several times [14,16], the ICs of ToMMV still remain to be constructed.

In the present study, we first report the full-length genome sequence of a tomato isolate for ToMMV from Liaoning Province in China. Next, an infectious cDNA clone was constructed with the full-length ToMMV genomic expressed downstream of the double 35S promoter (2 × 35S) by homologous recombination and was named as pToMMV. Additionally, the infectivity of this ToMMV isolate was tested on its hosts by *Agrobacterium tumefaciens*-mediated infiltration. Surprisingly, agroinfiltration of pToMMV enabled fast recovery of large amounts of virions from the agroinfiltrated leaves in the model host *N. benthamiana* and showed strong infection ability and developed severe symptoms for the natural hosts tomato and pepper, which allowed a great molecular characterization of ToMMV-LN.

## 2. Materials and Methods

### 2.1. Virus Sources and RNA Extraction

ToMMV-infected tomato (*Solanum lycopersicum* L.) samples were collected from Liaoning Province in China. Tomato samples showed yellowing, shrinking of leaves and necrosis of fruits. Total RNA was extracted from tomato leaves with RNAiso Reagent (TaKaRa, Dalian, China). Reverse transcription (RT) using PrimeScript^TM^ RT Reagent Kit with gDNA Eraser (TaKaRa, Dalian, China) was conducted following the manufacturer’s instructions.

### 2.2. Sequencing of Complete Nucleotide Sequence and Phylogenetic Analysis

Based on complete nucleotide sequences of the ToMMV isolate from Mexico (GenBank accession number KF477193) and other *Tobamovirus* in GenBank (ToMMV Hainan isolate MG171192, ToMMV HN isolate MH381817), three specific primer pairs (ToMMV-1F/ToMMV-1R, ToMMV-2F/ToMMV-2R and ToMMV-3F/ToMMV-3R; Appendix A) were designed for amplifying the full-length genome of ToMMV-LN. Each fragment was 2000 bp to 2300 bp in length, and overlapping areas between the adjacent two segments were designed to make sure the subsequent sequence could be assembled. The polymerase chain reaction (PCR) was performed using a high-fidelity polymerase PrimeSTAR^TM^ Max DNA Polymerase (TaKaRa, Dalian, China) in 50 μL reactions using 2.5 μL cDNA template from infected plants, 3 μL of each specific primer, 25 μL PrimeSTAR^TM^ Max DNA Polymerase, 16.5 μL ddH_2_O. PCR conditions were 1 min at 94 °C, followed by 33 cycles of 10 s at 98 °C, 15 s at 55 °C and 2 min at 72 °C, followed by a final cycle of 10 min at 72 °C. The PCR products were recovered by using a DNA gel recovery kit (Axygen, San Francisco, CA, USA) and were ligated into the pMD18-T vector (TaKaRa, Dalian, China), respectively. Then the recombinant vectors were transfected into Top 10 (Warbio, Nanjing, China) competent cells. Subsequently, colony PCR was performed with universal primer pairs M13F/M13R and the cloned PCR fragments were sequenced by General Biosystems Company (Anhui, China).

The full-length genomic sequences of ToMMV-LN (MN853592) were assembled with aforementioned three segments by using sequence assembly with default parameters in Seqman (DNASTAR) software. These genomic sequences were identified in the National Center for Biotechnology Information (NCBI) using a BLAST search, and nucleotide sequence data used in this study were obtained from the GenBank database. Nucleotide sequences were aligned using the Clustal W progressive alignment method, and the neighbor-joining phylogenetic tree was constructed by Kimura 2-parameter model with 1000 bootstrap replicates using MEGA 6.0 software. Nucleotide sequence data used in this study were obtained from the GenBank database: ToMMV isolate from Mexico (KF477193), ToMMV isolate from the Netherlands (MN654021), ToMMV isolate from Brazil (MH128145), ToMMV isolate from Spain (KU594507), ToMMV isolates from the United States (KP202857, KT810183, KX898033 and KX898034) and ToMMV isolates from China (MH381817, KR924950, KR824951 and MG171192).

### 2.3. Construction of Infectious Clone of ToMMV by Homologous Recombination Approach

To determine the pathogenicity of the ToMMV-LN isolate, we first constructed the full-length infectious cDNA clone of ToMMV-LN. Therefore, a specific primer pair (ToMMV-ssF and ToMMV-ssR; Appendix A) was designed for amplifying the full-length genome of ToMMV-LN. PCR was performed using PrimeSTAR^TM^ Max DNA Polymerase as aforementioned. PCR conditions were 1 min at 94 °C, followed by 33 cycles of 10 s at 98 °C, 15 s at 55 °C and 4 min 30 s at 72 °C, followed by a final cycle of 10 min at 72 °C. The products were recovered by using DNA gel recovery kit (Axygen, San Francisco, CA, USA). The purified PCR product of the full-length cDNA was introduced into the binary vector pCB301-2 × 35S-RZ-NOS [40] linearized by two restriction *Stu* I and *Sma* I (New England Biolabs, Beijing, China) to produce pToMMV using homologous recombination approach by ClonExpress^TM^ II One Step Cloning Kit (Vazyme Biotech, Nanjing, China) according to the manufacturer’s protocol. The ligated product was transfected into Top 10 (Warbio Biotech, Nanjing, China) competent cells and cultivated on Luria–Bertani (LB) plates with Kanamycin (50 mg/L) for 14 h at 37 °C condition. Subsequently, we performed colony PCR and restriction enzyme digestion, and the fragments of pToMMV clone were verified by General Biosystems Company (Anhui, China) sequencing.

### 2.4. Agroinfiltration of ToMMV Infectious cDNA Clone on Nicotiana benthamiana, Tomato and Pepper

The constructs pToMMV and empty vector pCB301 were electroporated into *A. tumefaciens* strain GV3101, respectively. *N. benthamiana*, tomato and pepper infection were performed as described before [41,42]. In brief, the recombinant colonies were grown at 28 °C, 220 rpm in LB medium supplemented with Kanamycin (50 mg/L) and Rifampicin (25 mg/L), then centrifuged at 8000× *g* for 3 min and resuspended at OD_600_ = 0.5 with agroinfiltration buffer (10 mM MgCl_2_; 10 mM MES, pH 5.6; 100 µM acetosyringone) and kept at room temperature for about 2 h before agroinfiltration. Agroinfiltrated plants were maintained under white fluorescent light (16 h light/8 h dark) at 26 °C constant temperature greenhouse.

To determine the infectivity of pToMMV clone on its model host and natural hosts, 4–6 fully expanded leaves of *N. benthamiana*, 3–5 fully expanded leaves of tomato (Moneymaker) and pepper (G26) were agroinfiltrated gently using 1 mL needleless syringes. *A. tumefaciens* culture harboring empty vector pCB301 was used as mock-inoculation control. *N. benthamiana*, tomato and pepper plants were agroinfiltrated with pToMMV or empty vector pCB301 for 3 seedlings as a group, respectively. The biological experiments were repeated 3 times with 3 repetitions for each group.

### 2.5. Detection of ToMMV by RT-PCR and Northern Blots

To confirm the infectivity of pToMMV in plants, we detected whether the ToMMV produced RNA transcripts in the pToMMV-infiltrated leaves or not. To this end, total RNAs were extracted from the pToMMV-infiltrated leaves of *N. benthamiana* at 1 day post-infiltration (dpi), 2 dpi and 4 dpi, with the pToMMV plasmid as positive control and healthy *N. benthamiana* plant as negative control. Then RT-PCR was performed using specific primers ToMMCP-F/ToMMCP-R (Appendix A). In addition, the systemically infected leaves of *N. benthamiana* at 10 dpi, tomato at 12 dpi and pepper at 15 dpi were collected to confirm the presence of ToMMV RNA in the pToMMV-infected plants, respectively. Total RNAs were extracted for RT-PCR detection by using specific primers ToMMCP-F/ToMMCP-R. The symptoms of agroinfiltrated plants were recorded and photographed, respectively. Northern blot hybridization analysis of total RNAs extracted from the systemically infected *N. benthamiana* at 10 dpi was carried out as described previously [41]. In brief, DIG-labeled probe specific for sense CP gene was synthesized using a DIG High Prime RNA labeling kit (Roche, Basel, Switzerland). The membrane blots were hybridized with the specific DIG-labeled probe, then processed using a DIG-High Prime Detection Starter Kit II (Roche) and the manufacturer’s protocol.

### 2.6. Virus Purification and Morphological Observation using Transmission Electron Microscopy (TEM)

Virus purification was performed as described in [43] and made several changes. In brief, 5 mL precooled 0.1 M phosphate buffer (PB), pH 7.2, with 0.01 M EDTA was added to 5 g ToMMV-infected *N. benthamiana* leaves followed by homogenization in a mortar. The homogenate was centrifuged at 8000× *g* for 20 min at 4 °C. The initial supernatant liquid was mixed with chloroform at the volume proportion of 4:1, and centrifuged at 10,000× *g* for 30 min at 4 °C. The resulting supernatant was mixed with PEG-6000 (4% *w*/*v*) and NaCl (0.1 M), then blended and centrifuged at 10,000× *g* for 15 min at 4 °C. The pellet was resuspended in 0.01 M PB (pH 7.2) at 2% of the volume of the initial supernatant liquid. Finally, the resuspended liquid was observed under transmission electron microscopy (TEM) H-7650 (College of Life Sciences, Nanjing Agriculture University, Nanjing, China).

## 3. Results

### 3.1. Genome Organization and Phylogenetic Analysis of ToMMV-LN

Three fragments were amplified and assembled (Appendix A). The full-length genome sequence of the ToMMV-LN isolate is 6399 nts long (MN853592), and consists of four open reading frames (ORFs), which were annotated according to their identity with those typically found in known *Tobamoviruses*. Pairwise alignment sequence comparison showed that the overall nucleotide sequence similarity between ToMMV-LN and the ToMMV isolates from Mexico (KF477193) and ToMMV-HN (MH381817) was 99.49% and 99.3%, respectively, whereas ToMMV-LN shared only 49.73–84.79% nucleotide sequence identity with other reported *Tobamoviruses* (Table 1). Additionally, alignment of the amino acid sequence revealed that the MP of ToMMV-LN shared the lowest identities (27.64–79.10%) with those of other *Tobamoviruses*, whilst the 183K of ToMMV-LN had a relatively high amino acid sequence similarity (45.73–94.68%) with those of other *Tobamoviruses* (Table 1). This suggests that the CP of ToMMV-LN has a closer evolutionary relationship with the *Tobamoviruses*. To characterize the relationship between ToMMV-LN and other ToMMV isolates, a phylogenetic tree based on the complete nucleotide sequence of ToMMV-LN and other ToMMV isolate sequences was constructed using the Neighbor-joining method with 1000 bootstrap replications by MEGA 6.0 software (Figure 1). Phylogenetic analysis resulted in ToMMV-LN clustered together with the ToMMV isolates from the United States (KP202857, KT810183, KX898033 and KX898034) and Mexico (KF477193) and China (KR824950, KR824951, MH381817 and MG171192), revealing very high similarities (98%).

### 3.2. Synthesis of ToMMV Infectious Clone and Infection Assays on N. benthamiana

To develop the infectious clone of ToMMV-LN isolate, the full-length cDNA sequence of ToMMV-LN was driven by the 2 × 35S promoter and inserted into the linearized binary vector pCB301 to generate pToMMV (Figure 2A and Appendix A). Restriction endonuclease reaction was performed to determine that the pToMMV plasmid was successfully constructed (Figure 2B). The model plant *N. benthamiana* was primarily used to investigate the biological role of ToMMV-LN, and the ability of the pToMMV to infect plants was assessed by *Agrobacterium*-mediated infiltration assays (Figure 2C). The target bands of approximately 500 bps were detected in agroinfiltrated leaves of *N. benthamiana* with pToMMV at 1, 2 and 4 dpi, whereas no viral RNA could be detected in healthy *N. benthamiana* plants (Appendix A). Since the total DNA from plants or cDNA of ToMMV transferred by agroinfiltration cannot be reverse transcribed in the RT processes, it was shown that the genomic RNAs of ToMMV-LN were successfully transcribed in agroinfiltrated leaves with pToMMV. Subsequently, the *N. benthamiana* plants inoculated with pToMMV showed foliar mottle in upper leaves at 7 dpi (Figure 2D), and leaf distortion, necrosis symptoms and stunting in the whole plants at 10 dpi, while lethal necrosis of the entire plants occurred within 15 dpi.

To further confirm that these phenotypes were induced by the pToMMV, RT-PCR detection was performed. It showed that the bands targeting ToMMV CP were detected in the positive control (the pToMMV plasmid) and systemically infected *N. benthamiana* leaves at 7 dpi, whereas no target band could be detected in healthy *N. benthamiana* and plants agroinfiltrated with pCB301 (Figure 2E). Agroinfiltrated *N. benthamiana* leaves were processed to acquire partially purified virion preparations and observed with a transmission electron microscope. To confirm the replication of the pToMMV, Northern blot analysis was also conducted. Systemically infected leaves from *N. benthamiana* agroinfiltrated with pToMMV showed the presence of gRNA of ToMMV CP (Figure 2F, lanes 2 and 3), whereas no viral RNA could be detected in plants agroinfiltrated with pCB301 (Figure 2F, lanes 1). The electron micrographs showed rigid rod-shaped ToMMV virions with particle dimensions of approximately 300 nm long and 18 nm diameter, typical characteristics of *Tobamovirus* (Figure 2G). All these cases verified that agroinfiltration of full-length cDNA clones of ToMMV-LN resulted in replication and virion formation in *N. benthamiana* leaves.

### 3.3. Pathogenicity and Infectivity on Tomato and Pepper Agroinfiltrated with pToMMV

As the ToMMV-LN was isolated from tomato, we also tested the infectivity of this clone on its natural host. For tomato plants, the progeny of ToMMV-LN from cDNA clones caused narrowing, crinkling at 12 dpi, distorting, interveinal yellowing of upper leaves, occasionally their stems and leaf veins changed from green to purple at 20 dpi, and later pToMMV caused enations along the veins on the undersides of leaves at plant bases and the whole tomato plants developed stunting at 30 dpi; these were typical symptoms like wild-type ToMMV-infected tomato plants collected from Liaoning (Figure 3A,C,D). Similarly, we also chose another natural host, pepper, for the infectivity test. Pepper plants agroinfiltrated with pToMMV showed irregular yellow mottle, shrinking from the bottom to the middle of apical leaves at an early stage about 14 dpi, subsequently the apical leaves developed brown necrosis at 21 dpi and severe mosaic at 35 dpi, latterly pToMMV caused leaf abscission and stunting symptoms of the whole plants at 35 dpi (Figure 4A,B,D).

The RT-PCR detection showed that the bands targeting ToMMV CP could be detected in the positive control pToMMV plasmid, agroinfiltrated tomato at 12 dpi and agroinfiltrated pepper at 14 dpi, while no target band could be detected in the negative control healthy tomato, pepper and those agroinfiltrated with pCB301 (Figure 3B and Figure 4C). As shown (Table 2), syringe inoculation was used for pToMMV, resulting in 100% infection in tomato and 40–47% infection in pepper. This demonstrated that the pToMMV resulted in abundant replication in tomato plants. Infectivity of the pToMMV on pepper was lower than on tomato.

Taken together, these results indicate that we have successfully constructed a full-length cDNA of ToMMV (pToMMV), and it has high pathogenicity and could efficiently infect the model host *N. benthamiana* and its natural hosts tomato and pepper.

## 4. Discussion

The ongoing spread of ToMMV threatens the output and quality of vegetable crops of subsistence farmers across China [11,12,44]. In this study, the full-length genome of ToMMV on tomato from Liaoning Province in China was amplified. Subsequently, we constructed the infectious cDNA clone pToMMV, which combines the full-length cDNA clone of ToMMV with a binary vector pCB301. Significantly, it has highly pathogenicity on three *Solanaceae* crops, including *N. benthamiana*, tomato and pepper. To the best of our knowledge, this is the first report on successfully constructing a reverse genetics system of ToMMV. Significantly, it is highly beneficial to study gene function by modifying the full-length infectious cDNA clone pToMMV in planta.

It is noteworthy that both in vitro viral RNA transcripts and agroinfiltration assays are usually used for constructing infectious cDNA clones. Since in vitro transcripts have been used for building infectious cDNA clones from *brome mosaic virus* (BMV) since 1984 [45], it has been used in many plant viruses. Agroinfiltration was known as “agroinfection” when it was used to infiltrate turnip plants with *A. tumefaciens* cells carrying binary plasmids, constructed by CaMV DNA with different vectors [46]. It has already been performed with many viruses such as *potato leaf roll virus* (PLRV), *beet western yellows virus* (BWYV), *lettuce infectious yellows virus* (LIYV) and *potato virus S* (PVS) on numerous plants [47,48,49,50]. Agroinfiltration coupled with targeted mutagenesis technologies will greatly enhance viral transcription in planta and insights will be obtained into virus replication, recombination, trafficking, symptom elicitation and virus–host interactions [49]. In the present study, we developed a stable infectious cDNA clone for delivering ToMMV to host plants in vitro by agroinfiltration. Accordingly, agroinfiltration of pToMMV in *N. benthamiana* plants enabled fast recovery of large amounts of virions from the agroinfiltrated leaves, which allowed better molecular characterization of ToMMV.

As for the pathogenicity of ToMMV, symptom development is heavily dependent on the infected hosts. A previous report showed that *N. benthamiana* plants show crinkling and yellowing on upper leaves at 4 dpi and serious necrosis symptoms on the apical shoot of seedlings at 6 dpi when mechanically inoculated with crude sap of ToMMV [51]. In our study, pToMMV induced lethal necrosis on agroinfiltrated *N. benthamiana* within 15 dpi (data not shown). This further suggests that the infectious cDNA clone pToMMV was successfully constructed and replicated severe necrosis symptoms on *N. benthamiana*. Notably, local and systemic lethal necrosis of *N. benthamiana* was due to plant virus pathogenicity determinants which play important roles in the silencing suppression or regulation of N-mediated immunity [52,53,54]. However, the proteins encoded by ToMMV for necrosis symptom development on *N. benthamiana* remain to be determined. The pathogenicity test of pToMMV on pepper plants showed mosaic, mottle and necrosis symptoms on top leaves (Figure 4), which coincided with the previous report [11]. The previous reports show that tomato plants infected by ToMMV showed foliar mottle, stunting, leaf distortion and necrosis symptoms [12], while sometimes exhibiting severe mosaic and tapering of the leaflet phenotypes [14]. In this study, tomato agroinfiltrated with pToMMV showed not only malformation and shrinking of the top leaves but also enations along the blade margin on the undersides of the leaves (Figure 3). This reveals that pToMMV can replicate the symptoms observed in naturally infected tomato plants. Remarkably, enations along the blade margin on the undersides of the leaves had never been found on ToMMV-infected tomato plants before; the mechanism of enations formation is worthy of further research.

ICs are powerful genetic tools for both basic and translational research on viruses of plant and animals, including human. The reverse genetic system based on an IC for *severe acute respiratory syndrome coronavirus 2* (SARS-CoV-2) can be used to rapidly engineer viruses with desired mutations to study the virus in vitro and in vivo [55]. Neonatal gnotobiotic pigs inoculated with the chimeric ICs, combining *porcine deltacoronavirus* (PDCoV) with *sparrow-origin deltacoronaviruses* (SpDCoV), developed an attenuated symptom with the loss of tropism for the pig intestine, indicating limited infectious potential for cross-species [56]. In addition, Hao et al., using the IC of *barley yellow dwarf virus*-*GAV* (BYDV-GAV), demonstrated that the P4 protein was involved in cell-to-cell movement and caused stunting symptoms in wheat [57]. These reports all shed light on the multiple functions of ICs, including generating various mutations and studying the pathogenic mechanism and gene functions of viruses. In addition, virus-induced gene silencing (VIGS) vectors, which generated a *Tobamovirus* from an IC of *sunn hemp mosaic virus* (SHMV), were effective both in the expression and in the silencing of a transgene green fluorescent protein (GFP) and in silencing of an endogenous gene phytoene desaturase (PDS) on *N. benthamiana* [58]. An NbPDS fragment inserted into an IC of *tobacco necrosis virus A* Chinese isolate (TNV-A(C)) as an inverted repeat produced a VIGS derivative that could silence endogenous NbPDS in *N. benthamiana* [59]. Therefore, for pToMMV, we can also use it to not only determine the pathogenic factor and study gene functions of ToMMV but also to construct the VIGS vectors used for gene silencing in planta. In summary, an infectious cDNA of ToMMV (pToMMV)was successfully constructed, and we chose agroinfiltration to demonstrate that pToMMV has high pathogenicity. This not only provides a powerful tool for us to investigate the characteristics of ToMMV, such as the gene functions or interactions between plant and virus and the insights into molecular mechanisms underlying viral infection, but also develops ToMMV-based vectors for gene silencing in host plants or expression of foreign proteins.

## Figures and Tables

**Figure 1 viruses-13-01050-f001:**
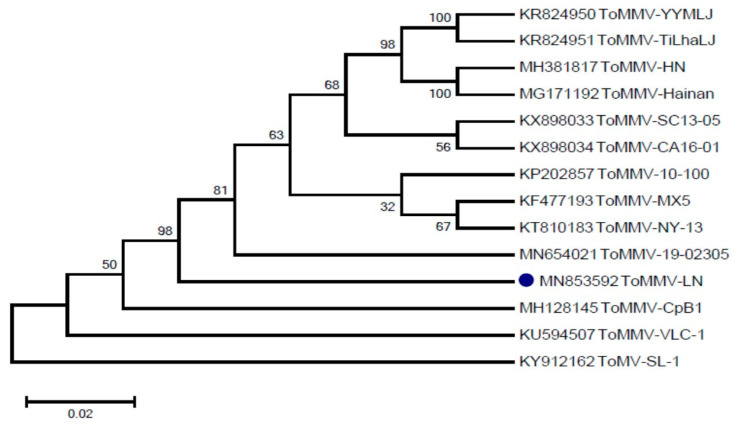
Phylogenetic tree based on the complete nucleotide sequence of ToMMV-LN and other ToMMV isolate sequences.

**Figure 2 viruses-13-01050-f002:**
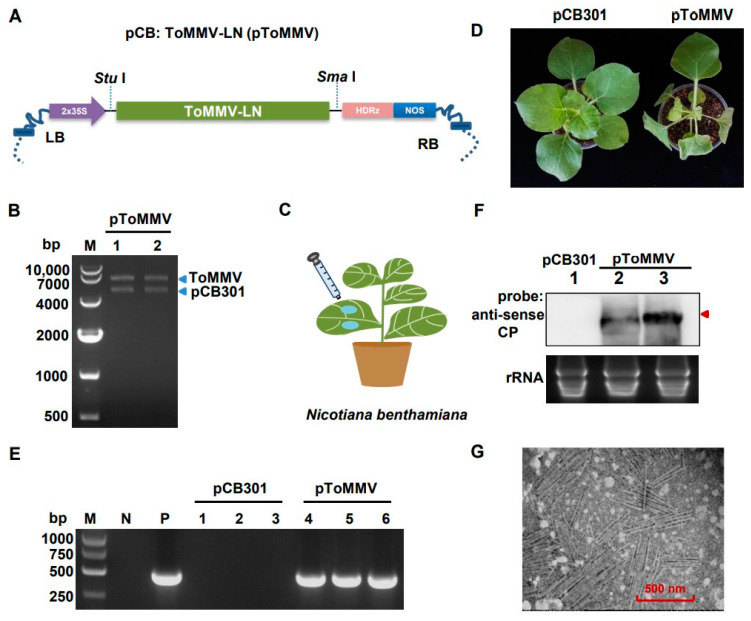
Transcription and replication of pToMMV in *N. benthamiana*. (**A**) Schematic diagram showing the infectious cDNA clone pCB: ToMMV-LN. 2 × 35S: CaMV 35S promoter. HDRz: hepatitis delta virus ribozyme. NOS: NOS terminator. LB: left border sequence. RB: Right border sequence. (**B**) *Agrobacterium*-mediated inoculation of pToMMV in *N. benthamiana* leaves. (**C**) Confirmation of the pToMMV infectious clone by restriction endonuclease digestion. M: DL 10,000 marker. Lanes 1-2: pToMMV plasmids digested with *Stu* I and *Sma* I. (**D**) Symptoms of *N. benthamiana* plants inoculated with pToMMV at 10 dpi. (**E**) RT-PCR detection to determine ToMMV infection in systemically infected leaves at 7 dpi. M: DL5000 DNA Marker. P: positive control, pToMMV plasmid. N: negative control, healthy *N. benthamiana* plant. Lanes 1–3, 4–6: *N. benthamiana* plants agroinfiltrated with pCB301 and pToMMV. (**F**) Northern blot analysis of total RNAs from the systemically infected leaves of *N. benthamiana* agroinfiltrated with pCB301 empty vector and pToMMV. Lane 1, 2–3: *N. benthamiana* plants agroinfiltrated with pCB301 and pToMMV. The total RNAs were detected using DIG-labeled anti-sense CP probe. Red arrow indicates genomic RNAs of ToMMV. Ethidium bromide staining was used as an RNA loading control. (**G**) Transmission electron micrograph for pToMMV resulted in abundant replication and virion formation in *N. benthamiana* leaves.

**Figure 3 viruses-13-01050-f003:**
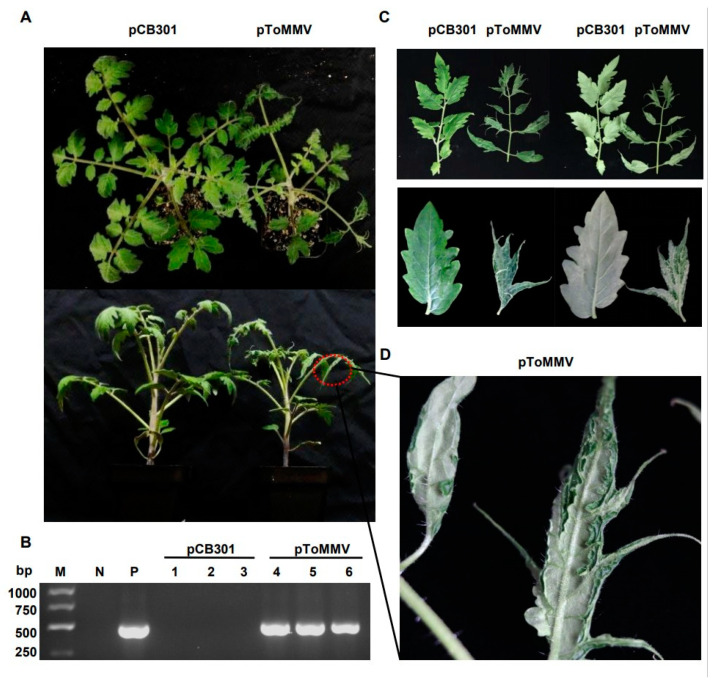
Plant phenotypes and ToMMV detection after agroinfiltration of pToMMV on tomato plants. (**A**) Symptoms of tomato plants agroinfiltrated with pToMMV at 30 dpi. (**B**) RT-PCR detection to determine ToMMV infection at 12 dpi. M: DL5000 Marker. N: healthy tomato plant. P: pToMMV plasmid. Lanes 1–3, 4–6: tomato plants agroinfiltrated with pCB301 and pToMMV. (**C**,**D**) The pToMMV caused enations along the veins on the undersides of leaves at plant bases on tomato plants.

**Figure 4 viruses-13-01050-f004:**
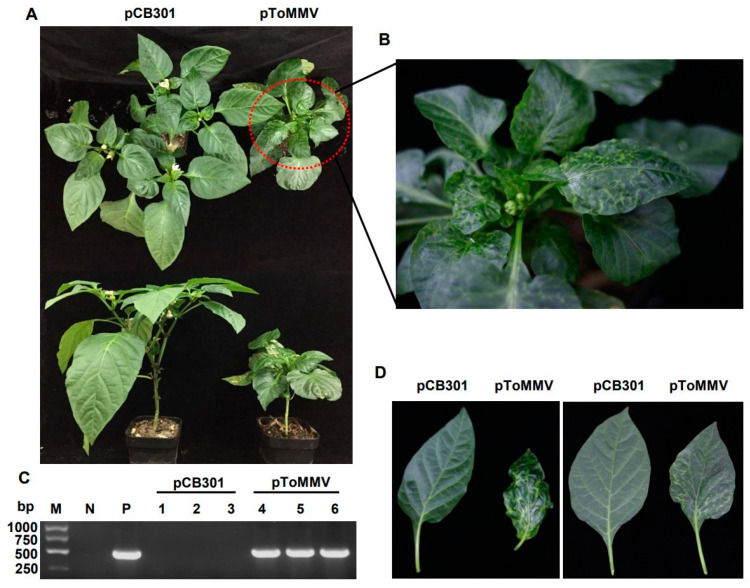
Plant phenotypes and ToMMV detection after agroinfiltration of pToMMV on pepper plants. (**A**) Symptoms of pepper plants agroinfiltrated with pToMMV at 35 dpi. (**B**) Partial enlarged view of pepper plant that was systemically infected with pToMMV. (**C**) RT-PCR detection to determine ToMMV infection at 14 dpi. M: DL5000 Marker. N: healthy pepper plant. P: pToMMV plasmid. Lanes 1–3, 4–6: pepper plants agroinfiltrated with pCB301 and pToMMV. (**D**) The pToMMV caused leaf abscission, severe mosaic on pepper leaves.

**Table 1 viruses-13-01050-t001:** Nucleotide and amino acid sequence identities (%) among ToMMV-LN and other ToMMV isolates and other *Tobamoviruses*.

Virus Name	Accession Number	Genome ^a^ (%)	126 K ^b^ (%)	183 K ^b^ (%)	MP ^b^ (%)	CP ^b^ (%)
ToMMV-MX5	KF477193	99.48	99.46	99.57	98.51	100
ToMMV-SC13-05	KX898033	99.36	99.37	99.50	98.51	100
ToMMV-NY-13	KT810183	99.34	99.37	99.50	98.88	99.37
ToMMV-YYMLJ	KR824950	99.34	99.73	99.75	98.88	100
ToMMV-10-100	KP202857	99.33	99.55	99.57	98.51	100
ToMMV-TiLhaLJ	KR824951	99.31	99.64	99.69	98.88	100
ToMMV-HN	MH381817	99.30	99.55	99.63	98.51	100
ToMMV-Hainan	MG171192	99.25	99.64	99.69	98.51	100
ToMMV-19-02305	MN654021	99.17	99.64	99.69	98.13	99.37
ToMMV-CpB1	MH128145	99.11	99.28	99.50	98.51	100
ToMMV-CA16-01	KX898034	99.09	99.64	99.69	98.51	100
ToMMV-VLC-1	KU594507	99.08	99.46	99.50	98.88	100
ToMV-SL-1	KY912162	84.79	94.27	94.68	79.10	91.19
ToMV-AH4	KU321698	84.63	94.09	94.62	78.73	91.82
TBRFV-Tom1-Jo	KT383474	80.64	91.58	92.20	71.27	86.16
RheMV-Henan	EF375551	78.14	89.43	90.47	70.52	82.50
PMMoV-pMG	KX063611	68.25	73.37	75.79	63.43	71.07
TMGMV-CaJO	MK648158	63.71	65.05	68.32	54.85	71.07
RMV-R14	HQ667979	59.26	60.25	64.32	34.94	48.43
CMoV	AB261167	50.78	39.09	44.89	27.64	41.98
CGMMV-SH	D12505	49.73	40.67	45.73	27.64	33.54

Note: ^a^ Nucleotide sequence and ^b^ amino acid sequences.

**Table 2 viruses-13-01050-t002:** Infection rate of syringe agroinfiltration with pToMMV in two different natural hosts.

Replicate	Syringe Agroinfiltration			
	No. of Tomato Plants Infected/Inoculated	Infection Rate (%)	No. of Pepper Plants Infected/Inoculated	Infection Rate (%)
I	15/15	100	6/15	40
Control	0/5	0	0/5	0
II	13/13	100	6/13	46.1
Control	0/5	0	0/5	0
III	15/15	100	7/15	46.7
Control	0/5	0	0/5	0

## Data Availability

Not applicable.

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
