# Peer review of "Synthesis and Characterization of a Full-Length Infectious cDNA Clone of *Tomato Mottle Mosaic Virus"

_viruses, 2021, doi:10.3390/v13061050_

Round 1

Reviewer 1 Report

This publication reports the cloning, characterization and construction of an infections clone for the Tomato Mottle Mosaic Virus. Such infectious clone provides a very useful tool for further studies of the viral infection cycle. The experimental approach is straight forward and sound. The paper is well written, provides detailed information regarding the experimental approach and the conclusions are supported by the data.

I do not have any major comments, but I have two minor questions that I would like to see detailed in the material and methods:

  1. i) what approach was used to obtain and verify the 5’ end of the newly characterized ToMMV?
  2. ii) in the material and methods it is indicated, in 2.2. and 2.3. that “PCR was performed following the manufacturer’s protocols”. Could you specify which kits were used and if a high fidelity polymerase was used.

Reviewer 2 Report

The virus nomenclature is often wrong. Authors should check the ICTV website for the correct nomenclature.

L45 and 48 both state ToMMV is an emerging virus. Once is sufficient.

How has the virus spread rapidly around the world. Seed-borne?

You claim reverse genetics is a 'practical technology' but don't define reverse genetics. The reference you give to describe reverse genetics in maize (Stadler, 1930) does not utilise viruses, as implied. There is nothing in the abstract to suggest the infectious clone was used for reverse genetics (RG). This suggests RG is a potential future use of this clone, and therefore a lengthy description of RG (lines 60-77) is not required here.

L78 I think you mean first full-length genome sequence.

L93 Which primers were used for RT?

L104 You claim the PCR products were transfected directly into E. coli cells. No mention is made of adding a bacterial origin of replication, so I cannot see how these 2.0 – 2.3 kb PCR amplicons replicated in the cells. Please clarify.

How were the three 2-2.3 kb fragments joined together? Fully describe.

L106 You mention ‘the recombinant clones’. What are these? There is no description of making recombinant clones.

How were sequence alignments done? Which models were assumed when doing the phylogenetic analysis. Note that the NJ method calculates pairwise ide3ntity, not phylogeny as claimed. If the authors wish to calculate phylogeny, they should use an appropriate algorithm, such as Mr Bayes or ML.

L123 How was the PCR of the full-length genome done. Details of substrate (ss cDNA, PCR products?) enzymes and amplification conditions should be provided.

The first paragraph of the results section describes in detail the architecture of the genome, which is almost identical to two isolates from Mexico and China, the first described in 2013. None of this detail is necessary because it has already been described by others.

Table 1 shows pairwise identities between ToMMV-LN and two other ToMMV isolates, whereas Fig 1 shows 12 other complete genome sequences. All sequenced ToMMV isolates should be shown in Table 1, not only two.

I agree that the authors have constructed an infectious clone. I feel they have provided adequate evidence.

L319. “In the present study … this is something of a milestone for the reverse genetics system construction for ToMMV” I am unsure how construction of the ToMMV infectious close was done by RG.

Also L345 

The virus nomenclature is often wrong. Authors should check the ICTV website for the correct nomenclature.

L45 and 48 both state ToMMV is an emerging virus. Once is sufficient.

How has the virus spread rapidly around the world. Seed-borne?

You claim reverse genetics is a 'practical technology' but don't define reverse genetics. The reference you give to describe reverse genetics in maize (Stadler, 1930) does not utilise viruses, as implied. There is nothing in the abstract to suggest the infectious clone was used for reverse genetics (RG). This suggests RG is a potential future use of this clone, and therefore a lengthy description of RG (lines 60-77) is not required here.

L78 I think you mean first full-length genome sequence.

L93 Which primers were used for RT?

L104 You claim the PCR products were transfected directly into E. coli cells. No mention is made of adding a bacterial origin of replication, so I cannot see how these 2.0 – 2.3 kb PCR amplicons replicated in the cells. Please clarify.

How were the three 2-2.3 kb fragments joined together? Fully describe.

L106 You mention ‘the recombinant clones’. What are these? There is no description of making recombinant clones.

How were sequence alignments done? Which models were assumed when doing the phylogenetic analysis. Note that the NJ method calculates pairwise ide3ntity, not phylogeny as claimed. If the authors wish to calculate phylogeny, they should use an appropriate algorithm, such as Mr Bayes or ML.

L123 How was the PCR of the full-length genome done. Details of substrate (ss cDNA, PCR products?) enzymes and amplification conditions should be provided.

The first paragraph of the results section describes in detail the architecture of the genome, which is almost identical to two isolates from Mexico and China, the first described in 2013. None of this detail is necessary because it has already been described by others.

Table 1 shows pairwise identities between ToMMV-LN and two other ToMMV isolates, whereas Fig 1 shows 12 other complete genome sequences. All sequenced ToMMV isolates should be shown in Table 1, not only two.

I agree that the authors have constructed an infectious clone. I feel they have provided adequate evidence.

L319. “In the present study … this is something of a milestone for the reverse genetics system construction for ToMMV” I am unsure how construction of the ToMMV infectious close was done by RG.

Likewise L345 "...the reverse genetics system of ToMMV was successfully constructed..."

In the Introduction I would have liked to have seen descriptions of all the other tobamovirus infectious clones made, and what they had been used for. This could have been discussed further in the Discussion in relation to your clone. The question ‘Why make the clone?” was not adequately answered. Could it be used in gene editing applications, for example?

Round 2

Reviewer 2 Report

The authors are congratulated on their work. They have largely applied my suggestions to improve the manuscript.

Some minor changes to the abstract are suggested below:

Delete 'security and quality' from the first sentence of thew abstract.

Replace 'by agroinfiltration with upon agroinfiltration in the abstract.

Replace amount with number in the abstract (L21).

Replace This is the first report that with This is the first report of

Replace We believe that described here will allow the studies of ToMMV genes function, pathogenicity and virus-host interaction with We believe that this infectious clone will enable further studies of ToMMV gene function, pathogenicity and virus-host interactions.

There remain minor grammatical errors throughout the manuscript. I recommend a thorough edit for scientific English if possible.
